# Comparative Analysis of the Phenolic Profile of *Lycium barbarum* L. Fruits from Different Regions in China

**DOI:** 10.3390/molecules27185842

**Published:** 2022-09-09

**Authors:** Wenwen Duan, Zhijie Zhang, Jingjing Zhu, Dong Zhang, Dan Qian, Fei Teng, Yifan Zhao, Fengming Chen, Raorao Li, Jin Yang

**Affiliations:** 1Institute of Chinese Materia Medica, China Academy of Chinese Medical Sciences, Beijing 100700, China; 2Experimental Research Centre of Chinese Medical Sciences, Beijing 100700, China; 3School of Chemistry and Chemical Engineering, North Minzu University, Yinchuan 750021, China; 4Key Laboratory for Chemical Engineering and Technology, State Ethnic Affairs Commission, Yinchuan 750021, China

**Keywords:** *Lycium barbarum* L. fruits, phenolic compounds, profile, classification

## Abstract

*Lycium barbarum* L. (LB) fruits have high nutritive values and therapeutic effects. The aim of this study was to comprehensively evaluate the differences in phenolic composition of LB fruits from different geographical regions. Different methods of characterization and statistical analysis of data showed that different geographic sources of China could be significantly separated from each other. The highest total phenolic compound (TPC) content was observed in LB fruits from Ningxia (LBN), followed by those from Gansu (LBG) and Qinghai (LBQ). The Fourier transform infrared (FTIR) spectra of LB fruits revealed that LBQ had a peak at 2972 cm^−1^ whereas there was no similar peak in LBG and LBQ. A new HPLC method was established for the simultaneous determination of 8 phenolic compounds by quantitative analysis of multiple components by a single marker (QAMS), including 4 phenolic acids (chlorogenic acid, caffeic acid, 4-hydroxycinnamic acid, and ferulic acid), 1 coumarin (scopoletin), and 3 flavonoids (kaempferol-3-O-rutinoside, rutin, and narcissoside). It was showed that rutin was the most dominant phenolic compound in LBQ, although the average content of 4 phenolic acids was also high in LBQ, and scopoletin was the richest in LBG. UHPLC-Q-TOF-MS was used to qualitatively analyze the phenolics, which showed LBN was abundant in phenolic acids, LBQ was rich in flavonoids, and coumarins were the most plentiful in LBG. In conclusion, this study can provide references for the quality control and evaluation of phenolics in LB fruits and their by-products.

## 1. Introduction

The plant genus Lycium (Solanaceae) is distributed widely in the world with high nutritional and medicinal values, and occurs in America, Africa, and Eurasia [1,2]. One of the most widely used species is *Lycium barbarum* L. (LB), which has been utilized as a commodity worldwide and has become a super food [2]. This species grows primarily in Asia and has also been cultivated in Europe and the Mediterranean. The main producing district of LB is China, particularly northwest China, where cultivation has a history of over 2000 years. LB is the most widely distributed cultivar in China [3]. The fruits of LB are used as a medicine named Goqizi, which can nourish the liver and kidney and replenish vital essence to improve eyesight [4]. Modern pharmacological studies on LB indicate that it has the beneficial effects of immune modulation, anti-aging effects, improving osteoblastic proliferation, radiological protection, ameliorating hepatic or brain injuries and impaired locomotor activities, neuroprotective effects, prevention of benign prostatic hyperplasia, delay of retinopathy, antioxidant and enzyme inhibitory effects, anti-inflammatory properties, anti-diabetes effects, anti-cancer properties, anti-hypertension effects, and cardioprotective effects [5,6,7,8,9,10,11,12,13,14,15,16,17]. LB has high nutritive value and therapeutic effects relevant to bioactive components, including polysaccharides, phenolics, carotenoids, alkaloids, vitamins, amino acids, and fatty acids [18,19,20,21].

In recent years, studies on phenolic compounds in LB have been second only to those on polysaccharides. Phenolics are aromatic rings with one or more hydroxyl groups that consist of simple phenols, polyphenols, benzoic and cinnamic acids, coumarins, tannins, lignins, lignans, and flavonoids [22,23]. These compounds are known as antioxidant and bioactive agents, with great benefits to health and in the prevention and treatment of diseases [24]. Therefore, researchers have paid increasing attention to the qualitative and quantitative analysis of phenolic compounds in LB, which mainly contain phenolic acids, flavonoids, phenolic amides, lignans, lignins, stilbenes, alkylphenols, curcuminoids, and terpenes. Among these compounds, phenolic acids and flavonoids are the best-studied constituents in LB fruits. The principal analysis methods utilize LC-MS, followed by HPLC and UHPLC. Thirty-five polyphenolic compounds were detected and quantified in the fresh and dried *Lycium* fruits, including five phenolic acids, 11 anthocyanins, and 19 phenolamides, using UHPLC-ESI-Q-TOF-MS [25]. After extraction by condensation reflux, hydrochloric acid acidification, and ethyl acetate, nine phenolic acids were determined in LB fruits by HPLC [26]. The isolation and purification of polyphenols from LB fruits were carried out by ultrasound-assisted extraction and solid-phase extraction, and then 10 phenolic acids and 11 flavonoids were identified and quantified by UHPLC-UV [27]. Furthermore, the quantitative analysis of complex components in herbs or foods is difficult in the absence of reference standards or the expensive cost of reference standards [28]. Consequently, a quantitative method with wide applicability should be established. Within the scope of a certain linearity range, the amount (weight or concentration) of one component is proportionate to the response values of the detector [29]. Quantitative analysis of multiple components by a single marker (QAMS) is a simple and economical method that only requires one standard reference, and all analytes in the sample can be identified simultaneously [30,31]. It was essential to select a suitable internal reference (IR) in order to establish the relative correction factor (RCF) between IR and other effective ingredients, and RCF can be influenced by many factors, such as laboratories, chromatographic instrument systems, packing, and the models of chromatographic columns [29,32]. The method of QAMS has been widely accepted and applied in the quality control of herbal medicine, which has been adopted by the Chinese Pharmacopoeia, the United States Pharmacopoeia, and the European Pharmacopoeia Standards. The study of simultaneous determination of phenolic compounds in LB fruits using HPLC-QAMS has been reported scarcely.

Ecological factors had a significant effect on fruit morphology and bioactive constituents. High soil, air temperatures, low altitude, light intensity, and moderate soil moisture were shown to be suitable conditions to produce Lycium fruits with a high content of nutritious metabolites [33]. Nzeuwa et al. [34] found that there was a slight difference in the contents of nutrients and phytochemicals among Lycium fruits from different areas, and the total phenol content of fruits grown in Nepal was higher than that of China. Lu et al. [35] reported that there are distinct differences in the functional components and antioxidant activity of *Lycium barbarum* L. fruits from different regions in China. Geographical factors had a great influence on phenolic compositions. Nevertheless, little is known about the overall distinction of phenolic compounds in LB fruits from different geographical sources.

In the present study, LB fruits coming from three major regions of China (Ningxia, Gansu, and Qinghai) were analyzed in multiple methods. The objective of this study was to evaluate the phenolic profile of LB fruits from different regions using qualitative and quantitative methods in order to gain a profound understanding of the phenolic diversity. We established a new method for quantifying eight phenolic compounds in LB fruits from different regions in China by HPLC combined with QAMS. The qualitative analysis of the phenolic profile was determined by UHPLC-Q-TOF-MS.

## 2. Results and Discussion

### 2.1. Physical Characteristics of LB Fruits

LB is widely cultivated in China, particularly in northwest districts. Generally, there were quite some different distinctions in fruit sizes and tastes of LB from different habitats [36]. A total of nine batches of samples were observed, including their color, shape, diameter and length (Figure 1). The appearance of all LB fruits was mostly red and fusiform. The highest values of diameter and length were observed for LBQ, which were 6.93 ± 0.18 and 17.10 ± 2.58 mm, respectively, while those of LBN were fractionally behind, with values of 6.67 ± 0.63 and 14.19 ± 1.24 mm. In the LBG fruits, the diameter was 5.66 ± 0.16 mm, and the length was 11.43 ± 2.09 mm. In a recent study, LB fruit morphological traits were also recorded from three regions, and the results showed that fruits from Qinghai were the largest, followed by those from Xinjiang and Ningxia. While the detailed morphological characteristics of LBN were smaller than those of LB fruits from other regions, it was traditionally an authentic (Daodi) herb in China [35].

### 2.2. TPC Content

The Folin–Ciocalteu assay was used for TPC content determinations. There were obvious differences in TPC contents in LB fruits from different regions. The highest TPC content was observed for LBN (29.931 ± 1.70 mg GAE/100 g), followed by LBQ (29.080 ± 1.08 mg GAE/100 g) and LBG (27.835 ± 3.11 mg GAE/100 g). According to the following scatter diagram (Figure 2), the TPC contents in LBN were higher than in LBG and LBQ, and the TPC contents of LBN and LBQ were more stable and consistent than LBG. The TPC content is influenced by many factors, including geographical, environmental, and cultivation methods [37]. Lu et al. also reported that the highest TPC contents were in LB fruits from Zhongning of Ningxia, which was greater than in samples from Gansu and Xinjiang [35].

### 2.3. FTIR-ATR

Fourier transform infrared spectroscopy is a widespread technique in the analysis of food components and can be a tool for rapid evaluation of foods and their by-products [38]. The spectra of the phenolic extracts of LBN, LBG and LBQ are shown in Figure 3. The spectra were dominated by typical vibrations in the OH region (3400–3200 cm^−1^) and aromatics (1500–1300 cm^−1^) related to phenolic compounds [39]. Peaks at 2923 and 2853 cm^−1^ were mainly associated with the hydrocarbon chains of the lipids or lignins [40]. The results showed that there was some difference in the 3000–2800 cm^−1^ region. Between them, the spectra of LBQ had two peaks at 2972 cm^−1^ (C-H stretching of the methylene bridges) and 2927 cm^−1^ (C-H stretching vibration) [41,42,43], and the absorbance of LBG and LBN was only at 2928 or 2927 cm^−1^, respectively. The absorption intensity of LBQ was higher than that of LBG and LBN at 2972 cm^−1^. A study from Peng et al. identified seven species and three variations of genus *Lycium* in China by FTIR, based on the additive infrared spectroscopy absorption of the chemical components and the differences of their relative contents in various Gouqi [44]. This method could provide a new way for the identification of LB fruits.

### 2.4. Method Validation and the Relative Correction Factor of HPLC-QAMS

All of the calibration curves and their linear regression equations of eight quantitative phenolic compounds including chlorogenic acid, caffeic acid, 4-hydroxycinnamic acid, ferulic acid, scopoletin, kaempferol-3-O-rutinoside, rutin, and narcissoside, are displayed in Table 1. The correlation coefficients (r) ranged from 0.9993 to 0.9999. The limit of detection (LOD) ranged from 0.0007 to 0.0094 μg/mL, and the limit of quantification (LOQ) from 0.0025 to 0.0314 μg/mL. The precision, repeatability, stability, and recovery of the eight analytes were presented in Table 1. The relative standard deviations (RSD) values of precision ranged from 0.07% to 2.38%. Six samples from the same batch were analyzed by an identical method, and the RSD values of repeatability were all lower than 2.96%. The RSD values of stability were less than 2.48. The mean recoveries of the eight analytes ranged from 95.94% to 104.36%, and the RSD values of recovery were under 3.00%.

To establish a new HPLC-QAMS method, some factors, such as columns and instruments, were required for detection. Other variables, such as wavelength, temperature, flow rate, and injection volume were also considered in order to gain an appropriate gradient elution method. During the research, it was found that these factors had effects on peak number, peak shape, and retention time. In this study, the influences of different instruments and chromatographic columns on relative correction factor (RCF) were investigated (Table 2). The results proved that different instruments and columns had no significant effects on the RCF value.

### 2.5. Quantitative Determination of Phenolic Compounds in LB Fruits from Different Regions

Generally, quantitative methods usually use multiple standards to determine analytes. QAMS only requires the use of an IR to detect all analytes [28]. In a recent study, a method for simultaneous determination of four carotenoids in *Lycium barbarum* was built by using QAMS [45]. In this study, a new HPLC-QAMS method was established (Figure 4A) that could be used to determine eight phenolic compounds (Figure 4B) in LB fruits. The analytes included four phenolic acids (chlorogenic acid, caffeic acid, 4-hydroxycinnamic acid, and ferulic acid), one coumarin (scopoletin), and three flavonoids (kaempferol-3-O-rutinoside, rutin, and narcissoside). Among them, scopoletin was selected as the IR with its moderate retention time, stable property, low price, and its peak shape that were presented well. Compared with external standard methods (ESM) that were used for comparison, the contents of the other seven analytes by the QAMS method showed a narrow gap. Their average RSD values were less than 5.0% (Table 3). The results showed that there was no significant difference between the results of the ESM and QAMS methods, and it was indicated that the establishment of HPLC-QAMS was feasible for the determination of eight phenolic compounds in LB fruits by using scopoletin as IR.

The highest mean contents of chlorogenic acid, caffeic acid, 4-hydroxycinnamic acid, ferulic acid, and rutin were in LBQ, which were 0.0068 mg/g, 0.0071 μg/g, 0.0097 μg/g, 0.0043 μg/g and 0.0196 μg/g, respectively. The highest mean contents of scopoletin, narcissoside, and kaempferol-3-O-rutinoside were 0.0035, 0.0011 and 0.0009 μg/g in LBG, respectively. There was a significant difference among the regions.

For phenolic acids, 4-hydroxycinnamic acid was the main phenolic acid in LBQ, which accounted for 0.0097 μg/g. The next was caffeic acid with a content of 0.0071 μg/g, and the lowest was ferulic acid with 0.0043 μg/g. These differences were observed in the heatmap (Figure 5). The changes in phenolic acids in LBN were the same as those in LBQ, and the highest content of 4-hydroxycinnamic acid was 0.0043 μg/g, and the lowest was 0.0034 μg/g. The conditions were particularly clear in LBG. The minimum and maximum contents were 0.0071 μg/g of 4-hydroxycinnamic acid and 0.0017 μg/g of ferulic acid. The greatest total phenolic acid content was 0.0279 μg/g in LBQ, which was 1.75 times and 1.73 times higher than that in LBN and LBG, respectively.

For flavonoids, rutin had the highest content in all samples, which was 0.0196 μg/g in LBQ, 1.56 times and 1.48 times compared with LBN and LBQ, respectively. There were nearly no differences in the contents of narcissoside and kaempferol-3-O-rutinoside applied to all samples. The sum of the flavonoid contents was 0.0143, 0.0152 and 0.0215 μg/g in LBN, LBG and LBQ, respectively. LBG has the greatest content of scopoletin at 0.0037 μg/g, and that in LBN was similar to that in LBQ.

In general, the total content of four phenolic acids was less than that of three flavonoids. The highest total content of the eight analytes was observed for LBQ, followed by LBG and LBN. The maximum contents of scopoletin, kaempferol-3-O-rutinoside and narcissoside were observed in LBG. The contents of other analytes were the highest in LBQ.

### 2.6. Qualitative Analysis of Phenolic Compounds in LB Fruits by UPLC-Q-TOF-MS

By analyzing mass data from previous literature and studies [46,47,48,49,50,51,52,53,54,55,56,57,58,59,60,61,62,63,64,65,66,67], 74 phenolic constituents were identified in total in our samples, including 18 flavonoids, 19 phenolic acids, seven phenolic amides, six coumarins, three terpenes, three chromenes, two lignans and 16 other phenolics. The data were presented in Table 4. There were 46, 50 and 43 phenolic compounds identified in this study in LBN, LBG and LBQ fruits, respectively (Figure 6A). Among them, 26 phenolic substances were found in all LB fruits from three different regions. In particular, 11 phenolic compounds were unique in LBN, including six phenolic acids, one terpene and four other phenolics. In addition, 16 phenolic compounds were only identified in LBG, including one flavonoid, one phenolic acid, two coumarins, two phenolic amides, one terpene, one chromene, one lignan and seven other phenolics. Meanwhile, eight phenolic compounds barely existed in LBQ, including three flavonoids, one coumarin, one phenolic amide and three other phenolics.

The main phenolic compounds in LB fruits are phenolic acids and flavonoids, and it is also crucial to learn about their amounts and varieties in these medicinal fruits [27,68]. As shown in Figure 6B, there were 18, 12, and 11 and 11, 13, and 16 phenolic acids and flavonoids in LBN, LBG and LBQ fruits, respectively. LBN was rich in phenolic acids, and LBQ was rich in flavonoids. The amounts of coumarins and phenolic amides in LBG were greater than those in LBQ and LBN.

Principal component analysis (PCA) is a mathematical tool that aims to represent the variation present in the dataset using a small number of factors [69]. It is used to identify how one sample differs from another, which variables contribute most to the difference, and whether these variables are correlated [70]. Cossignani et al. found that the geographic origin of goji samples could be discriminated against using PCA for fatty acids and sterol percent compositions [71]. In a recent study by Gong et al., samples of *Lycium barbarum* L. from the same place could be partially discriminated by PCA using stable isotopes, earth elements, free amino acids, and saccharides [72]. To obtain the overall characteristics and similarities of phenolic compounds in LB fruits from three different regions, a PCA test based on identified 74 phenolic compounds was performed in this study. The two main principal components accounted for approximately 62.7% of the total variance. The results showed that in the PCA model (Figure 7), the LB fruits could be differentiated into three groups which contained LBN, LBQ, and LBG respectively.

In a study from Poland [73], it was observed that Goji fruit (*Lycium barbarum* L.) from China showed a wide variety of available phenolic acids using chromatographic analysis (LC-ESI-MS/MS). Phenolic acids, coumaric, isoferulic, and caffeic acids, and their derivatives, were found to be the dominant ones of *Lycium barbarum* cultivated in Greece [74]. Phenolic acids were determined as the most abundant compounds of *Lycium barbarum* L. cultivated in Italy, followed by flavanols [75]. There were significant differences in the numbers and types of phenolics in LB fruits from three different regions in China, indicating that regions were important factors in the quality of LB fruits. The results also showed that LB fruits were abundant in phenolic compounds and had great potential as natural functional foods and nutritional pharmaceutic.

## 3. Materials and Methods

### 3.1. Materials and Chemicals

LB fruits (Cultivar: Ningqi 7) from three different regions (15 batches of Ningxia, 15 batches of Gansu and 8 batches of Qinghai) in China were collected at harvest from places of origin (Ningxia: 35°14′–39°23′N, 104°17′–107°39′ E, Gansu: 32°31′–42°57′ N, 92°13′–108°46′ E, Qinghai: 31°4′–39°19′ N, 89°35′–103°03′ E). The samples were preserved at −40 °C and then freeze-dried under vacuum. Whole LB fruits including LBN, LBG, and LBQ were ground into a fine powder and stored at −40 °C.

Chlorogenic acid (≥96.1%), caffeic acid (≥99.7%), 4-hydroxycinnamic acid (≥99.7%), scopoletin (≥99.7%), ferulic acid (≥99.4%), kaempferol-3-O-rutinoside (≥94.0%) and rutin (≥91.6%) were obtained from the National Institutes for Food and Drug Control (Beijing, China). Narcissoside (≥98%) was purchased from Shenzhen Botaier Biotechnology Company (Shenzhen, China). HPLC-grade methanol and formic acid were purchased from Fisher and Roe Scientific Inc. Pure water was obtained from Wahaha Group Co., Ltd. (Hangzhou, China).

### 3.2. The Appearance Character of LB Fruits

The main physical characteristics of LB fruits from three different regions in China were observed. Due to the limited quantity of some sample batches, three batches of samples were selected for each region because of their abundant quantities. The color and shape of LB fruits were recorded, and the values of length and diameter were analyzed.

### 3.3. Extraction of Phenolic Compounds

Briefly, 1.5 g of dried powder of LB fruits was extracted with 10 mL of methanol/water solution (80:20, *v*/*v*) and subjected to ultrasound-assisted extraction for 30 min. The supernatant was obtained after filtration, and then the solutions were filtered through 0.22 μm microporous membranes and stored at −40 °C. Each sample was analyzed in duplicate. The filtrates were concentrated in a rotary evaporator at 45 °C and dried by vacuum freeze-drying for FTIR analysis. All 38 batches of samples were extracted and further analyzed for 3.4, 3.5, and 3.6.

### 3.4. Determination of the Total Phenolic Content

The TPC content of the extracts was determined by the Folin–Ciocalteu method [76], with slight modifications. A total of 1 mL of diluted extract was transferred to a 25 mL volumetric flask and mixed with 1 mL of Folin–Ciocalteu reagent and 2 mL of sodium carbonate (1 mol/L). Then, the solution was diluted with pure water to volume. Subsequently, the mixtures were incubated in darkness for 1 h. The absorbance was measured utilizing a UV-vis spectrophotometer (T6, Persee) at 760 nm against a blank. Each sample was tested in triplicate. Gallic acid was used as a standard to prepare the calibration curve. The results were expressed as milligram equivalents of gallic acid per 100 g (mg GAE·100 g^−1^) dry weight.

### 3.5. FTIR-ATR Analysis

The FTIR spectra were recorded on an FTIR spectrometer (PerkinElmer Frontier) equipped with a ZnSe crystal cell for attenuated total reflection (ATR) operation. The spectra were acquired (three scans per sample) in the midinfrared region of 4000–550 cm^−1^ at a resolution of 4 cm^−1^.

### 3.6. Analysis of Phenolic Composition by HPLC-QAMS

#### 3.6.1. Investigation of the Instrumental Conditions

HPLC analysis was performed on a Shimadzu HPLC-DAD (SIL-20A, SPD-M20A, CTO-20A) system and a Waters 2695 system, by using a Shimadzu GIST C18-AQ column (4.6 mm × 250 mm, 5 µm), a Welch Ulimate^®^ AQ-C18 column (4.6 mm × 250 mm, 5 µm), and a Kromasil column (4.6 mm × 250 mm, 5 µm). The mobile phases were A (methanol) and B (0.5% formic acid) at a flow rate of 1.0 mL/min. The gradient elution was as follows: 0–10 min, 2–20% A; 10–55 min, 20–25% A; 55–80 min, 25–30% A; 80–90 min, 30–40% A; 90–100 min, 40–45% A; and 100–110 min, 45–50% A. The injection volume was 20 μL. The column temperature was 35 °C, and the detection wavelength was 360 nm.

#### 3.6.2. Method Validation and Calculation of the Relative Correction Factor

Eight standards, including chlorogenic acid, caffeic acid, 4-hydroxycinnamic acid, scopoletin, ferulic acid, rutin, kaempferol-3-O-rutinoside and narcissoside, were prepared by dissolving them in methanol. The mixed standard solution was diluted to different concentrations and stored at 4 °C. The HPLC-QAMS method was validated in terms of precision, stability, reproducibility, linearity, LOD, LOQ and recovery. The linearity was established with the peak areas of six different concentrations for each phenolic compound. The LOD and LOQ were calculated at the signal-to-noise ratio of 3:1 and 10:1, respectively. The inter-day precision was evaluated by RSD under six repeated injections, which was assessed by repeatability. The repeatability was determined by analyzing six prepared repeated samples from the same batch. Recovery tests were measured by spiking six samples with known content which were determined in the repeatability tests from the same batch, with known amounts of each analyte. Scopoletin was applied as the internal reference (IR). The RCF was calculated according to the following formulas:f_s/i_ = (A_s_ × C_i_)/(A_i_ × C_s_)
where A_s_ and C_s_ represented the peak areas and concentrations of the IR, respectively, and A_i_ and C_i_ represented the peak areas and concentrations of analytes, respectively.

### 3.7. Qualitative Analysis of Phenolic Compounds by UHPLC-Q-TOF-MS

A total of 9 batches of samples were determined, which were the same as 2.1. Separation was performed on an Acquity UPLC BEH C18 column (2.1 mm×100 mm, 1.7 μm, Waters) using an Acquity UPLC system (Waters) with a column temperature of 35 °C. A volume of 2 μL was injected at a flow rate of 0.3 mL/min. The mobile phases were 0.1% formic acid (A) and methanol (B), and the gradient program was as follows: 0–0.5 min, 5% B; 0.5–2.0 min, 5–10% B; 2.0–4.5 min, 10–15% B; 4.5–7.0 min, 15–20% B; 7.0–20.0 min, 20–25% B; 20.0–32.0 min, 25–30% B; 32.0–35.0 min, 30–40% B; and 35.0–40.0 min, 40–50% B.

Analysis was performed on a UHPLC-Q-TOF-MSE (Waters) system equipped with a Xevo G2-S Q-TOF mass spectrometer, a lock-spray interface and an electrospray ionization (ESI) source operated in both positive and negative ionization modes. The capillary and cone voltages were 2.2 kV and 40 V, respectively. The temperature of the ionization source was 120 °C, and the ion collision energy was 20–50 eV. The mass range was set at 100–1800 *m*/*z*. The data were collected and processed by MassLynx4.1 software.

### 3.8. Statistical Analysis

All results were expressed as the mean, standard deviation (SD) and relative standard deviation (RSD). The data were analyzed using Microsoft Excel 2016, Origin 2021, SPSS Statistics 17.0, and SIMCA 14.1. The heatmap was generated using a bioinformatics network (http://www.bioinformatics.com.cn, accessed on 7 April 2022).

## 4. Conclusions

In this study, multiple analytical methods were applied to reveal the differences between LB fruits from the three different regions. While LB fruits from Qinghai were the largest in the size, the highest TPC content was observed in LBN. In the FTIR spectra, a similar trend was found in LBN and LBG, whereas there was a special peak in LBQ. It was shown that rutin was the main constituent in LB fruits with a new HPLC-QAMS method, especially in LBQ. The distribution of phenolic compounds was determined by UHPLC-Q-TOF-MS analysis. The most amounts of phenolic acids were in LBN, flavonoids in LBQ, and coumarins in LBG. In our study, the qualitative and quantitative phenolic compound profiles of LB fruits from different regions in China were established. The results will be helpful for the quality control and evaluation of LB fruits and their by-products as functional foods.

## Figures and Tables

**Figure 1 molecules-27-05842-f001:**
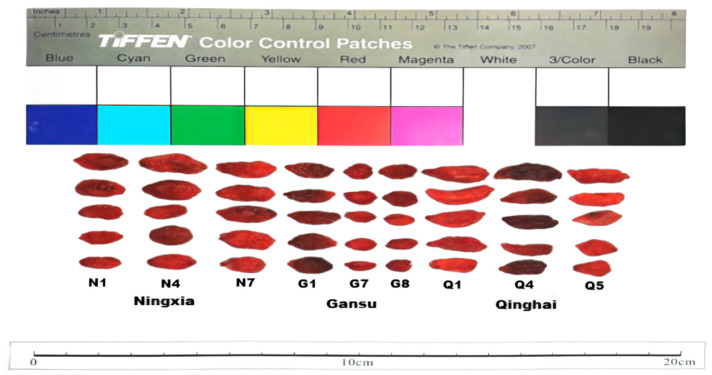
The characteristics of LB fruits from different regions.

**Figure 2 molecules-27-05842-f002:**
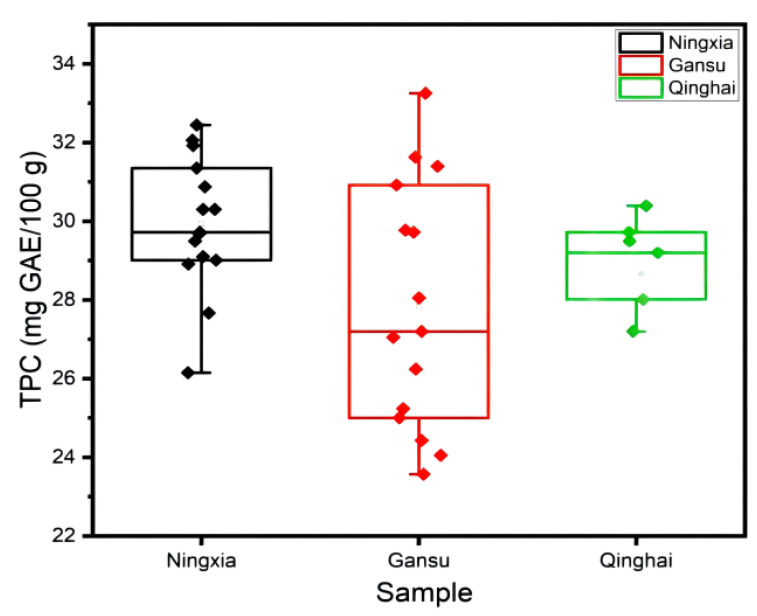
Total phenolic compound (TPC) content of LB fruits from different regions.

**Figure 3 molecules-27-05842-f003:**
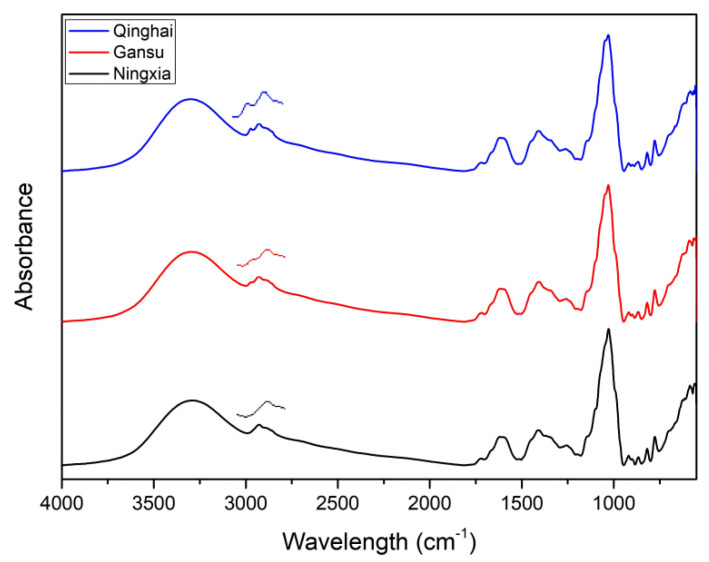
Spectra of phenolic extracts in LB fruits from different regions.

**Figure 4 molecules-27-05842-f004:**
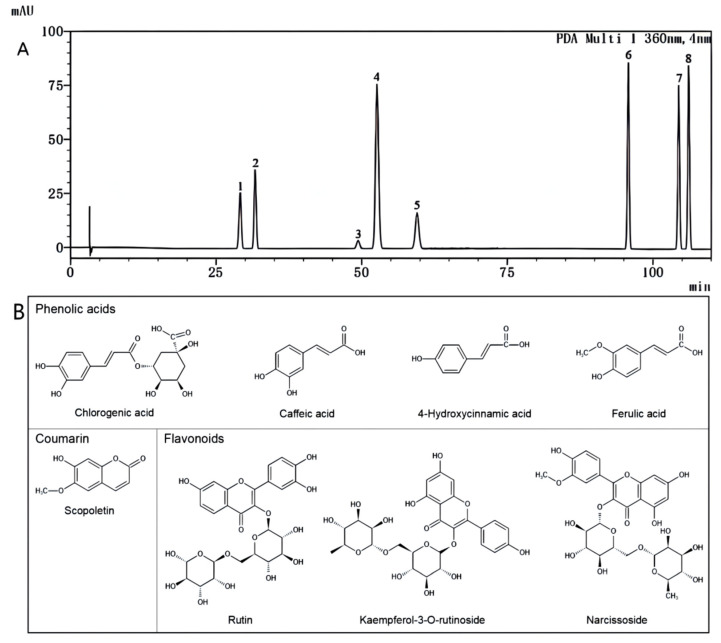
(**A**) HPLC chromatogram of mixed standard solutions (1: chlorogenic acid; 2: caffeic acid; 3: 4-hydroxycinnamic acid; 4: scopoletin; 5: ferulic acid; 6: rutin; 7: kaempferol-3-O-rutinoside; 8: narcissoside). (**B**) Chemical structures of eight phenolic compounds that were quantified in LB fruits.

**Figure 5 molecules-27-05842-f005:**
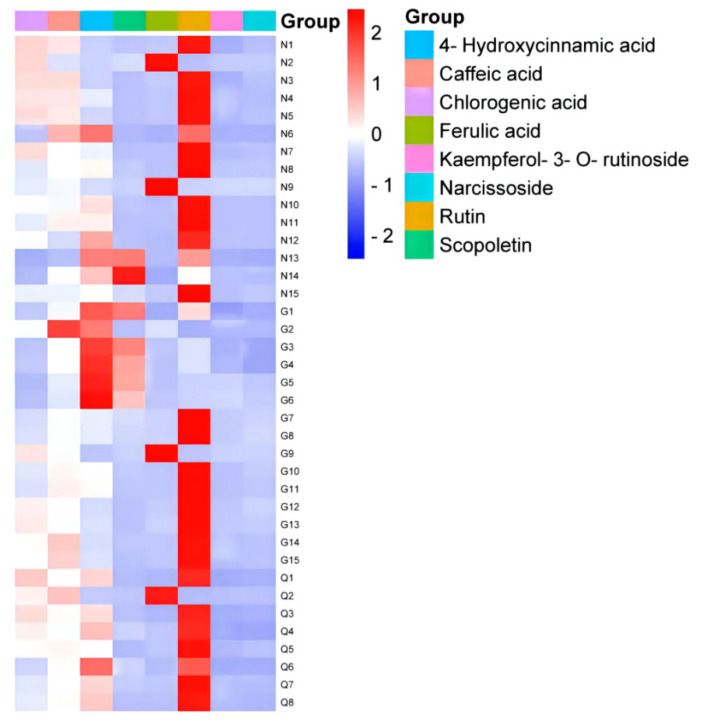
Heatmap of phenolic content in LB fruits from different regions.

**Figure 6 molecules-27-05842-f006:**
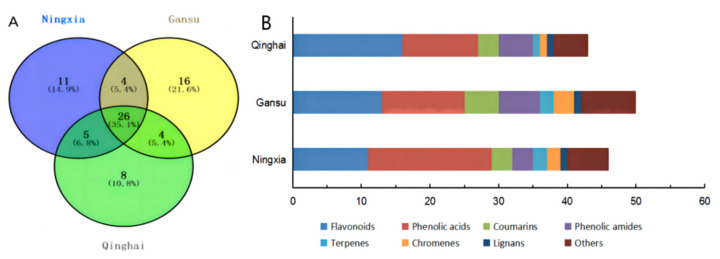
Qualitative analysis of phenolic compounds in LB fruits by UPLC-Q-TOF-MS. (**A**) The Venn diagram of phenolic compounds in LB fruits from different regions. (**B**) The types of phenolic compounds in LB fruits from different regions.

**Figure 7 molecules-27-05842-f007:**
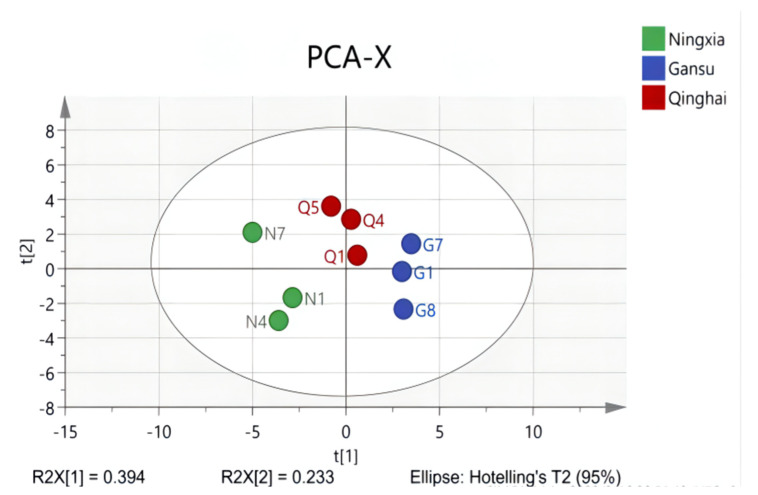
The PCA−X plot of LB fruits from different regions on the basis of 74 phenolic compounds identified by UPLC-Q-TOF-MS.

**Table 1 molecules-27-05842-t001:** Method validation data of 8 phenolic compounds by HPLC.

Reference Substance	Linearity			LOD (μg/mL)	LOQ (μg/mL)	Precision RSD (%)	Repeatability RSD (%)	Stability RSD (%)	Recovery	
	Regression Equation	Range (μg/mL)	r						Mean (%)	RSD (%)
Chlorogenic acid	Y = 14333X − 3436.9	0.39–24.95	0.9994	0.0019	0.0065	0.28	0.95	0.70	97.87	2.18
Caffeic acid	Y = 17526X − 2502.4	0.81–25.90	0.9999	0.0040	0.0135	0.15	1.80	0.22	100.25	2.20
4-Hydroxycinnamic acid	Y = 2081.6X − 615.94	1.88–30.15	0.9998	0.0094	0.0314	2.38	1.10	2.28	103.56	3.00
Scopoletin	Y = 48845X − 4441.1	0.42–27.10	0.9999	0.0021	0.0071	0.86	2.10	0.86	103.72	2.26
Ferulic acid	Y = 13278X − 2424.1	0.64–10.16	0.9993	0.0032	0.0106	1.95	2.96	2.48	104.36	2.34
Rutin	Y = 32673X − 6917.5	1.68–215.20	0.9999	0.0021	0.0070	0.07	1.13	0.49	100.23	1.25
Kaempferol-3-O- rutinoside	Y = 35325X − 1081.2	0.15–9.72	0.9999	0.0008	0.0025	0.71	2.81	0.84	95.94	0.99
Narcissoside	Y = 39925X − 1172.1	0.15–9.46	0.9999	0.0007	0.0025	1.28	2.14	1.29	101.63	1.03

**Table 2 molecules-27-05842-t002:** The value of RCF of each component in different influence factors.

Instrument	Column	f_a/d_	f_b/d_	f_c/d_	f_e/d_	f_f/d_	f_g/d_	f_h/d_
SHIMADZU-LC-20AD	Shim-pack GIST C18-AQ	0.285	0.343	0.041	0.258	0.658	0.664	0.760
	Welch Ulimate^®^ AQ-C18	0.287	0.341	0.041	0.261	0.647	0.657	0.754
	Kromasil	0.286	0.342	0.041	0.260	0.653	0.660	0.757
Waters2695	Shim-pack GIST C18-AQ	0.286	0.336	0.041	0.260	0.654	0.672	0.775
	Welch Ulimate^®^ AQ-C18	0.277	0.322	0.039	0.256	0.646	0.658	0.767
	Kromasil	0.282	0.325	0.039	0.262	0.649	0.657	0.757
Mean		0.284	0.335	0.041	0.260	0.651	0.661	0.761
RSD (%)		1.365	2.766	2.936	0.814	0.718	0.860	1.023

a. Chlorogenic acid; b. Caffeic acid; c. 4-Hydroxycinnamic acid; d. Scopoletin; e. Ferulic acid; f. Rutin; g. Kaempferol-3-O-rutinoside; h. Narcissoside.

**Table 3 molecules-27-05842-t003:** The average content of eight phenolic compounds in LB fruits from three different regions (μg/g).

Compounds	LBN			LBG			LBQ		
	ESM	QAMS	RSD (%)	ESM	QAMS	RSD (%)	ESM	QAMS	RSD (%)
Chlorogenic acid	0.0041	0.0039	2.79	0.0025	0.0026	3.68	0.0068	0.0071	2.89
Caffeic acid	0.0041	0.0040	1.60	0.0048	0.0047	1.02	0.0071	0.0070	1.26
4-Hydroxycinnamic acid	0.0043	0.0045	3.12	0.0071	0.0068	3.27	0.0097	0.0093	2.85
Scopoletin	0.0022	-	-	0.0037	-	-	0.0022	-	-
Ferulic acid	0.0034	0.0036	3.31	0.0017	0.0018	2.52	0.0043	0.0046	4.19
Rutin	0.0126	0.0127	0.39	0.0132	0.0131	0.42	0.0196	0.0196	0.05
Kaempferol-3-O-rutinoside	0.0008	0.0008	0.43	0.0009	0.0009	0.39	0.0009	0.0009	0.38
Narcissoside	0.0009	0.0009	0.46	0.0011	0.0011	0.28	0.0010	0.0010	0.45

**Table 4 molecules-27-05842-t004:** The qualitative analysis of phenolics in LB fruits from different regions in China.

No.	t_R_ (min)	Identification	Formula	Mass (*m*/*z*)	Cacl. Mass (*m*/*z*)	mDa	Fragements (MS^2^)	LBN	LBG	LBQ
**Flavonoids**										
1	2.54	2’-hydroxyflavanone	C_15_H_12_O_3_	239.0662	239.0708	−4.6	191.0190; 130.0862; 124.0392	√	√	√
2	3.13	7-methoxy-2-phenyl-3,4-dihydro-2H-1-benzopyran-4-one	C_16_H_14_O_3_	253.0823	253.0865	−4.2	218.0653; 194.9445; 137.0233; 128.0341	√	√	
3	4.16	3-(2,4-dihydroxybenzoyl)-4,5- dimethyl-5-[4-methyl-5-(4-methyl-5- (4-methyl-2-furyl)-3(E)-penten -1-l-yl]tetrahydro-2-furanone	C_24_H_28_O_6_	411.177	411.1808	−3.8	249.1241; 135.0446		√	√
4	5.58	4-(2-Carboxyethenyl)-2-(3,4-di hydroxy phenyl)-2,3-dihydro-7- hydroxy-3-methylester, [2α, 3β, 4(E)-3-benzofuran carboxylic acid	C_16_H_20_O_10_	371.0973	371.0978	−0.5	163.0396; 119.0497		√	√
5	8.61	Quercetin-rhamno-tri-hexoside	C_39_H_50_O_26_	933.2519	933.2512	0.7	470.2283			√
6	9.95	Quecetin 3-O-galactosylrutinoside	C_33_H_40_O_21_	771.2003	771.1984	1.9	609.1465; 301.0349	√	√	√
7	11.43	Quercetin3-O-α-L-rhamno pyranosyl-(1→6)-β-D-galacto pyranosyl-7-O-β-D-sophoroside	C_39_H_50_O_26_	933.2511	933.2115	−0.1	609.1454; 301.0342	√	√	√
8	12.61	Quercetin deoxyhexose -hexose- deoxyhexose	C_33_H_40_O_20_	755.2018	755.2035	−1.7	593.1491			√
9	13.77	5,4′-dihydroxy-3′-methoxyflavonol-3-O-glucosyl-(1→6)-glucosyl-7-O rhamnoside	C_34_H_42_O_21_	785.2155	785.214	1.5	623.1630; 315.0510		√	√
10	15.71	Chakaflavonoside A	C_39_H_50_O_25_	917.2571	917.2563	0.8	194.9445			√
11	16.15	Qucercetin 3-O-glucosylrutinoside	C_33_H_40_O_21_	771.1995	771.1984	1.1	609.1457; 301.0345	√		√
12	17.78	Quercetin 3-O-rutinoside-(1-2)-O- rhamnoside	C_33_H_40_O_20_	755.2018	755.2035	−1.7	300.0262; 194.9411	√	√	√
13	20.42	Parviside A	C_39_H_50_O_26_	933.2516	933.2512	0.4	771.1987; 292.9211	√		√
14	21.43	Sachaloside IV	C_33_H_40_O_21_	771.1985	771.1984	0.1	301.0340; 194.9416	√	√	√
15	23.46	Rutin	C_27_H_30_O_16_	609.1459	609.1456	0.3	300.0270; 101.0231	√	√	√
16	30.71	Kaempferol 3-O-rutinoside	C_27_H_30_O_15_	593.1505	593.1506	−0.1	285.0395; 194.9424	√	√	√
17	32.92	Isorhamnetin 3-O-rutinoside	C_28_H_32_O_16_	623.1620	623.1612	0.8	315.0498; 194.9424	√	√	√
18	38.27	Swertianolin	C_20_H_20_O_11_	435.0919	435.0927	−0.8	216.9271; 194.9447		√	
**Phenolic acids**										
19	2.61	Quinic acid derivate	C_11_H_22_O_9_	297.1182	297.1186	−0.4	239.0646; 191.0183; 163.0382; 124.0394	√		
20	4.43	Caffeic acid derivative	C_13_H_32_O_14_	411.1747	411.1714	3.3	179.0345; 161.0244; 135.0441	√		
21	4.56	Dicaffeoylquinic acid derivative	C_22_H_30_O_15_	533.1498	533.1506	−0.8	515.1372; 191.0541; 163.0391; 135.0437; 109.0285	√	√	
22	4.90	Caffeoylquinic acid derivative 1	C_34_H_36_O_19_	747.1754	747.1773	−1.9	191.0555; 179.0542; 163.0398; 161.0452	√		√
23	5.15	Coumarinylquinic acid derivative 1	C_34_H_36_O_19_	747.1756	747.1773	−1.7	191.0555; 163.0399; 145.0294; 119.0498	√		
24	5.39	3-O-(4’-O-Caffeoyl glucosyl)quinic acid	C_22_H_28_O_14_	515.1401	515.1401	0	353.0868; 191.0559; 163.0396; 135.0441	√	√	√
25	5.53	Caffeoylquinic acid derivative 2	C_18_H_24_O_14_	463.1114	463.1088	2.6	203.0826; 191.0560	√	√	√
26	5.92	Coumarinylquinic acid derivative 2	C_34_H_36_O_19_	747.1749	747.1773	−2.4	163.0397; 145.0290; 119.0492	√	√	√
27	6.03	Coumarinylquinic acid derivative 3	C_35_H_38_O_20_	777.1855	777.1878	−2.3	461.1659; 193.0501; 113.0236	√	√	√
28	6.20	Feruloylquinic acid derivative 1	C_39_H_64_O_20_	851.3942	851.3913	2.9	337.0762; 216.9280; 193.0505; 191.0553; 163.0391	√	√	√
29	6.92	Feruloylquinic acid derivative 2	C_28_H_38_O_20_	693.187	693.1878	−0.8	337.0762; 216.9270; 191.0348; 163.0393	√		
30	7.09	Coumarinylquinic acid derivative 4	C_20_H_38_O_22_	629.1824	629.1776	4.8	337.0756; 179.0342; 163.0390; 161.0237	√		
31	7.22	5-O-(3’-O-Caffeoyl glucosyl)quinic acid	C_22_H_28_O_14_	515.14	515.1401	−0.1	323.0767; 191.0557; 179.0348; 161.0241; 108.0201	√	√	√
32	7.40	Caffeoylquinic acid derivative 3	C_33_H_54_O_15_	689.3394	689.3384	1	191.0556; 179.0349; 163.0395; 135.0445	√		
33	7.64	Chlorogenic acid	C_16_H_18_O_9_	353.087	353.0873	−0.3	191.0555	√	√	√
34	9.46	Caffeic acid	C_9_H_8_O_4_	179.1572	179.157	0.2	135.0447	√	√	√
35	11.3	p-hydroxycinnamic acid	C_9_H_8_O_3_	163.0396	163.0395	0.1	135.0441; 119.0496	√	√	√
36	13.89	Ferulic acid	C_10_H_10_O_4_	193.1847	193.184	0.7	145.0321	√	√	√
37	18.11	Clinopodic acid Q	C_33_H_32_O_17_	699.1534	699.1561	−2.7	194.9423		√	
**Coumarins**										
38	0.72	Cephalosol	C_16_H_14_O_8_	333.0583	333.061	−2.7	260.8785; 128.9590; 112.9853		√	
39	7.26	Umbelliferone	C_9_H_6_O_3_	163.0402	163.0395	0.7	127.0398			√
40	7.36	Esculetin	C_9_H_6_O_4_	177.0191	177.0188	0.3	163.0393; 135.0443; 119.0494		√	
41	8.96	(R)-6-hydroxymellein diglycoside	C_21_H_28_O_13_	487.1449	487.1452	−0.3	470.2279; 163.0393; 145.0290	√	√	√
42	11.77	Scopoletin	C_10_H_8_O_4_	193.0506	193.0501	0.5	163.0400; 133.0293	√	√	√
43	15.77	6,7-di-O-(2′, 3′, 4′, 6′-tetra-O- acetyl-β-D-galactopyranosyl)-4- methylcoumarin	C_38_H_44_O_22_	851.2272	851.2246	2.6	623.1628; 292.9214; 194.9420; 191.0555	√	√	
**Phenolic amides**										
44	31.60	N-feruloyltiramine	C_18_H_19_NO_4_	312.1234	312.1236	−0.2	292.9217; 194.9418; 178.0499; 148.0522; 135.0443	√	√	√
45	7.65	Caffeoyl (dihydrocaffeoyl) spermidine-tri-hexose	C_43_H_63_N_3_O_21_	956.3878	956.3876	0.2	677.1931; 470.2284; 191.0549			√
46	8.21	Lycibarbarspermidine S	C_37_H_53_N_3_O_16_	794.3357	794.3348	0.9	632.2822; 470.2292; 334.1768	√	√	√
47	9.88	Lycibarbarspermidine P	C_31_H_43_N_3_O_11_	632.2831	632.2819	1.2	470.2290; 334.1764	√	√	√
48	10.25	Lycibarbarspermidine R	C_31_H_43_N_3_O_11_	632.2823	632.2819	0.4	540.2342; 470.2287; 334.1765; 135.0446		√	√
49	11.04	(E)-3-(3,4-dihydroxyphenyl)-N- ethylacrylamide	C_11_H_13_NO_3_	206.082	206.0817	0.3	194.9426; 135.0445		√	
50	11.18	N,N’-dicaffeoylspermidine	C_25_H_31_N_3_O_6_	470.2281	470.2291	−1	220.0976; 163.0396		√	
**Terpenes**										
51	0.82	Vaccihein A	C_18_H_18_O_9_	377.085	377.0873	−2.3	341.1085; 191.0560; 179.0556; 101.0238	√		
52	5.08	Mudanpioside J	C_31_H_34_O_14_	629.1833	629.187	−3.7	163.0397; 135.0444	√	√	√
53	6.80	Mudanpioside J isomer	C_31_H_34_O_14_	629.1836	629.187	−3.4	529.3028		√	
**Chromenes**										
54	13.02	2-(2-hydroxy-benzylidene)-3,3a dihydrocyclopenta [b] chromen -1(2H)-one	C_19_H_14_O_3_	291.099	291.1021	−3.1	159.0926; 130.0659	√	√	√
55	16.73	4,7-Dihydroxy-2-oxo-2H- chromene-3-acetyle derivative	C_25_H_31_N_3_O_5_	454.2336	454.2342	−0.6	163.0396		√	
56	22.33	4H-1-benzopyran-4-one,2-(3,4 -dimeth oxyphenyl)-6,8-di -β-D -glucopyranosyl-5,7-dihydroxy	C_27_H_30_O_16_	609.1445	609.1456	−1.1	300.0260; 194.9420	√	√	
**Lignans**										
57	37.47	Pharsyringaresinol	C_30_H_39_O_14_	623.2389	623.234	4.9	460.1757		√	
58	38.61	Terminaloside G	C_30_H_40_O_14_	623.2384	623.234	4.4	460.1749; 216.9263; 194.9442	√		√
**Others**										
59	0.76	Pentacenehydroquinone	C_22_H_14_O	293.0984	293.0966	1.8	215.0323; 131.0457		√	
60	2.77	2-(4-methoxyphenyl)-3,4-dihydro-2H-1-benzopyran-4-one	C_16_H_14_O_3_	253.0826	253.0865	−3.9	231.0291; 128.0351		√	
61	4.71	Caffeoyl derivative	C_34_H_36_O_19_	747.1744	747.1773	−2.9	629.1830; 487.1455; 163.0396			√
62	6.56	Levodopa	C_9_H_11_NO_4_	196.0612	196.061	0.2	161.0246; 122.0608	√		√
63	7.09	Juglanoside D	C_16_H_20_O_9_	355.1024	355.1029	−0.5	193.0500; 134.0366		√	
64	7.51	1-O-(E)-caffeoyl-β-D-glucopyranosyl-(1→2)-[β-D-glucopyranosyl-(1→6)]-β-D-glucopyranose	C_27_H_38_O_19_	665.1922	665.1929	−0.7	503.1395; 341.0870; 179.0348; 161.0241	√		
65	11.12	Rhinacanthin	C_25_H_30_O_5_	411.2125	411.2171	−4.6	163.0393			√
66	11.21	Kankanoside F	C_26_H_40_O_17_	623.2191	623.2187	0.4	468.2128; 332.1606		√	
67	12.54	2,3-diphenylphenol	C_18_H_14_O	245.0927	245.0966	−3.9	203.0822; 135.0447; 116.0498	√		
68	12.61	3-O-β-D-Apiofuranosyl(1→2)-β-D-glucopyranosyl Rhamnocitrin 4′-O -β-D-Glucopyranoside	C_33_H_40_O_20_	755.2043	755.2035	0.8	593.1512		√	
69	13.34	3,5-diphenylphenol	C_18_H_14_O	245.0934	245.0966	−3.2	203.0821; 135.0442	√	√	√
70	14.22	Verbascoside	C_29_H_36_O_15_	623.1973	623.1976	−0.3	461.1648; 194.9418; 161.0241	√		
71	15.25	1-O-[(5-O-syringoyl)-β-D-apiofuranosyl]-(1→2)-β-D-glucopyranosie	C_28_H_34_O_17_	641.1713	641.1718	−0.5	479.1167; 194.9416; 167.0342		√	
72	15.32	Lamiuside C	C_35_H_46_O_20_	785.2505	785.2504	0.1	771.1986; 194.9423; 161.0239	√		
73	19.61	(E)-2-({[2-(1,3-dioxan-2-yl) phenyl]imino}methyl)phenol	C_17_H_17_NO_3_	282.1133	282.113	0.3	194.9414		√	
74	37.45	Dihydroxy-3:5:3’:5’-tetra-2”- hydroxybenzyl-diphenylmethane	C_41_H_36_O_6_	623.2391	623.2343	−4.3	196.8947			√

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
