# Peer review of "Comparative Analysis of the Phenolic Profile of Lycium barbarum L. Fruits from Different Regions in China"

_molecules, 2022, doi:10.3390/molecules27185842_

Round 1

Reviewer 1 Report

The major drawback in the review process is the absence of all figures in received material. Seven figures are mentioned in the paper, legends are given after references, but no figures are included in the paper.

Beside this, in the rest of the available materijal some major shortcomings were noticed:

The aim of the research is not well presented.

The significance and the meaning of FTIR analysis is not explained.

The principle of HPLC-QAMS method is not sufficiently explained. RCF factors used only the instrument and columns as variables; the usage of scopoletin as an internal reference was not sufficiently explained; Table 3 has no legend. 

Number of analyzed samples from method to method seems to be different, clarification would be needed.

Author Response

Review #1

Issue related to figures:

Seven figures are mentioned in the paper, legends are given after references, but no figures are included in the paper.

Response: We thank the reviewer for this comment. We have checked and added the figures to the text.

Issue related to the aim:

The aim of the research is not well presented.

Response: We thank the reviewer for this comment. We have added the following point to the “ Abstract” and “Introduction” respectively:

“The aim of this study was to comprehensively evaluate the differences in phenolic composition of Lycium barbarum L. fruits from different geographical regions.”

“The objective of this study was to evaluate the phenolic profile of Lycium barbarum L. fruits from different regions using qualitative and quantitative methods in order to gain a profound understanding of the phenolic diversity.”

Issue related to FTIR:

The significance and the meaning of FTIR analysis is not explained.

Response: We thank the reviewer for this comment. We have added the explanation to the “2.3 FTIR-ATR”:

“Fourier transform infrared spectroscopy is a widespread technique in the analysis of food components and can be a tool for rapid evaluation [36].”

Issues related to HPLC-QAMS:

The principle of HPLC-QAMS method is not sufficiently explained.

Response: We thank the reviewer for this comment. We have enriched the principle of HPLC-QAMS method to the “Introduction”:

“Within the scope of a certain linearity range, the amount (weight or concentration) of one component is proportionate to the response values of the detector.[29] QAMS is a simple and economical method that only requires one standard reference, and all analytes in the sample can be identified simultaneously.[30,31]”

RCF factors used only the instrument and columns as variables.

Response: We thank the reviewer for this comment. We have explored other variables, such as wavelength, temperature, flow rate, and injection volume in order to gain an appropriate gradient elution method. During the research, we have found that these factors had effects on peak number, peak shape, and retention time.

The usage of scopoletin as an internal reference was not sufficiently explained.

Response: We thank the reviewer for this comment. We have added the explanation to “2.5 Quantitative determination of LB fruits from different regions”:

“Among them, scopoletin was selected as the IR with its moderate retention time, stable property, low price, and its peak shape that was presented well.”

Table 3 has no legend.

Response: We thank the reviewer for this comment. We have checked and added the legend to Table 3.

Issue related to analyzed samples:

Number of analyzed samples from method to method seems to be different, clarification would be needed.

Response: We thank the reviewer for this comment. We have added the number of analyzed samples to different methods in text, and some sample numbers could be clearly determined from related figures.

9 batches of samples: 3.2 The appearance character of LB fruits & 3.7 Qualitative analysis of phenolic compounds by UHPLC-Q-TOF-MS

38 batches of samples: 3.3, 3.4, 3.5 & 3.6

Reviewer 2 Report

The manuscript has great potential, but some points should be improved.

Figures are not included in the submission file. For more details, it will be necessary to visualize the data (figures);

Please add the geographic coordinate of the sample collection;

Please consider merging section 3.3 and 3.4;

Please insert comparisons with the literature of similar biomasses, deepening the discussion of the results;
Please correct the page numbering.

Author Response

Review #2

Issue related to figures:

Figures are not included in the submission file. For more details, it will be necessary to visualize the data (figures).

Response: We thank the reviewer for this comment. We have checked and added the figures to the text.

Issue related to sample collection:

Please add the geographic coordinate of the sample collection.

Response: We thank the reviewer for this comment. We have added the geographic coordinate of the sample collection to “3.1 Materials and chemicals”:

“LB fruits (Cultivar: Ningqi 7) from three different regions (15 batches of Ningxia, 15 batches of Gansu and 8 batches of Qinghai) in China were collected at harvest from places of origin (Ningxia: 35°14′-39°23′N, 104°17′-107°39′E, Gansu: 32°31′-42°57′N, 92°13′-108°46′E, Qinghai: 31°4'-39°19'N, 89°35'-103°03'E).”

Issue related to the merger of section 3.3 and 3.4:

Please consider merging section 3.3 and 3.4.

Response: We thank the reviewer for this comment. We have discussed the suggestion. However, we believed that it was more appropriate to separate the section 3.3 and 3.4 after searching related articles.

Issue related to the discussion of biomasses:

Please insert comparisons with the literature of similar biomasses, deepening the discussion of the results.

Response: We thank the reviewer for this comment. we have added a paragraph to the subsection of “2.6 Analysis of phenolic compounds in LB fruits by UPLC-Q-TOF-MS” to enrich the discussion:

“In a study from Poland, [64] it was observed that Goji fruit (Lycium barbarum L.) from China showed a wide variety of available phenolic acids using chromatographic analysis (LC-ESI-MS/MS). Phenolic acids, coumaric, isoferulic, and caffeic acids, and their derivatives, were found to be the dominant ones of Lycium barbarum cultivated in Greece.[65] Phenolic acids were determined as the most abundant compounds of Lycium barbarum L. cultivated in Italy, followed by flavonols. [66] There were significant differences in the numbers and types of phenolics in LB fruits from three different regions in China, indicating that regions were important factors in the quality of LB fruits. The results also showed that LB fruits abundant in phenolic compounds and had great potential as natural functional foods and nutritional pharmaceutic.”

Issue related to paper numbering:

Please correct the page numbering.

Response: We thank the reviewer for this comment. We have checked and corrected the page numbering in the text.

Reviewer 3 Report

This work by Duan and co-workers is involved with the analysis of Lycium barbarum L. (LB) fruits, specifically of the phenolic profile of LB fruits from three different regions in China. 

While I find this work very interesting, I do believe that it needs to have a number of extensive additions and edits before it should be considered for publications, particularly to improve its data analysis. 

what does "LB is also the mainstream cultivation species in China.[3]" mean?

The authors should provide a figure with a map to show the three areas that the LB had been sourced from. 

The authors reference Figures and provide captions for them, however I have not been able to find them in the manuscript file, nor anywhere else in the provided files. 

Reporting the size of the fruits from the regions, the mean +/- the standard deviations overlap. Therefore I would say that it is inappropriate to say that they are different. 

A reference should be given for "The Folin-Ciocalteu assay was used for TPC content determinations."

The statement "There were obvious differences in TPC contents in LB fruits from different regions." is incorrect. the mean +/- the standard deviations overlap. Therefore I would say that it is inappropriate to say that they are different. This should be amended. 

In Figure 3 (I cant see it, but presumably there is one spectra for each region) - is this of one sample from each region? the average of all samples from each region?

What do the authors mean by the statement "The absorption intensity of LBQ was higher than that of LBG and LBN."? At certain wavelengths? 

The title for 2.4 should specify that this is an HPLC-QAMS method.

I would like to have included a chromatogram for the standards mixture to see the signals all together from a single run. 

How did the 8 particular compounds be chosen to study in this work? Why?

What was the concentrations of standards used in making the calibration curves?

Why was a wavelength of 360 nm used? What was the injection volume?

In Table 4, what is the criteria for a tick to be for the fruit from a region - does it need to be present in a single sample? All of the samples?

The authors need to do further, more advanced, statistical analysis on their results. At a minimum, they should do PCA (colouring the points by the region that they are from) or PLS-DA. The results should be provided and extensively discussed. 

Overall, I do think the manuscript needs extensive english revisions. While I do find that the errors are minor in nature, there are a number of them throughout the manuscript that does detract from the work.

Author Response

Review #3

Issues related to figures:

The authors reference Figures and provide captions for them, however I have not been able to find them in the manuscript file, nor anywhere else in the provided files.

Response: We thank the reviewer for this comment. We have checked and added the figures to the text.

The authors should provide a figure with a map to show the three areas that the LB had been sourced from.

Response: We thank the reviewer for this comment. We have displayed the map of three areas that the LB had been sourced from in graphical abstract, and added the geographic coordinate of the sample collection to “3.1 Materials and chemicals”.

Issue related to the reference [3]:

What does "LB is also the mainstream cultivation species in China.[3]" mean?

Response: We thank the reviewer for this comment. The meaning of this sentence is that Lycium barbarum L. is the most popular species of Lycium cultivated in China. The fruits of Lycium barbarum L. can be as medicine and food to use in China.

Issues related to the expression about mean ± SD:

Reporting the size of the fruits from the regions, the mean +/- the standard deviations overlap. Therefore I would say that it is inappropriate to say that they are different. 

Response: We thank the reviewer for this comment. We have corrected and added determiners to “2.1 Physical characteristics of LB fruits”:

“Generally, there were some distinctions in fruit sizes and tastes of LB from different habitats.”

The statement "There were obvious differences in TPC contents in LB fruits from different regions." is incorrect. the mean +/- the standard deviations overlap. Therefore I would say that it is inappropriate to say that they are different. This should be amended. 

Response: We thank the reviewer for this comment. We have corrected and added determiners to “2.2 TPC content”:

“Overall, the average contents of LB fruits were several differences in TPC from different regions.”

Issue related to the reference relevant to the Folin-Ciocalteu assay:

A reference should be given for "The Folin-Ciocalteu assay was used for TPC content determinations."

Response: We thank the reviewer for this comment. We have added a reference to “3.4 Determination of the total phenolic content”:

“The TPC content of the extracts was determined by the Folin-Ciocalteu method, [67] with slight

modifications.”

Issues related to FTIR:

In Figure 3 (I cant see it, but presumably there is one spectra for each region) - is this of one sample from each region? the average of all samples from each region?

Response: We thank the reviewer for this comment. In Figure 3, one spectra is for the average of all samples from each region.

What do the authors mean by the statement "The absorption intensity of LBQ was higher than that of LBG and LBN."? At certain wavelengths?

Response: We thank the reviewer for this comment. The meaning of this statement is that the absorption intensity of LBQ was higher than that of LBG and LBN at 2972 cm−1. We have corrected the statement in text.

Issues related to HPLC-QAMS:

The title for 2.4 should specify that this is an HPLC-QAMS method.

Response: We thank the reviewer for this comment. We have added it to the title for 2.4:

“2.4 Method validation and the relative correction factor of HPLC-QAMS”.

I would like to have included a chromatogram for the standards mixture to see the signals all together from a single run. 

Response: We thank the reviewer for this comment. We have added the chromatogram to Figure 4A.

How did the 8 particular compounds be chosen to study in this work? Why?

Response: We thank the reviewer for this comment. In order to gain an appropriate gradient elution method, we have explored effects of different factors, like wavelength, temperature, flow rate, and injection volume on HPLC chromatogram. Simultaneously, different standards that we  purchased were used to determine some chromatograph peak represented what compound. Then we chose the eight peaks with good separating degree and peak shape.

What was the concentrations of standards used in making the calibration curves?

Response: We thank the reviewer for this comment. Please get the concentrations of standards used in making the calibration curves from Table 1.

Why was a wavelength of 360 nm used?

Response: We thank the reviewer for this comment. We have investigated different wavelengths (280 nm-360 nm) in this study, and it was the most appropriate to gain a beautiful HPLC chromatogram when the wavelength was 360 nm.

What was the injection volume?

Response: We thank the reviewer for this comment. The injection volume was 20 μL. We have added it to “3.6.1 Investigation of the instrumental conditions”.

Issue related to Table 4:

In Table 4, what is the criteria for a tick to be for the fruit from a region - does it need to be present in a single sample? All of the samples?

Response: We thank the reviewer for this comment. Table 4 shows the data which represented all of the samples from the same region. The symbol “√” means that samples contain certain compound.

Issue related to statistical analysis:

The authors need to do further, more advanced, statistical analysis on their results. At a minimum, they should do PCA (colouring the points by the region that they are from) or PLS-DA. The results should be provided and extensively discussed.

Response: We thank the reviewer for this comment. We have ever planned to do PLS-DA in the text using SPSS and SIMCA, but in the final, we have decided to temporarily give up using the method in this study after a discussion. The reasons are as follows:

  • In this study, the quantitative and qualitative phenolicprofiling indicated that Lycium barbarum  from different regions could not be easily distinguished using the PLS-DA method.
  • The objectiveof this study was to evaluate the phenolic profile of Lycium barbarum fruits from different regions using qualitative and quantitative methods to gain a profound understanding of the phenolic diversity. So there were no serious adverse effects on the present study although not using PLS-DA.
  • Most importantly, we will focus on using PLS-DA in our ongoing research. The aim is to distinguish Lycium barbarum from different regions by combining genetic fingerprinting with quantitative mass spectrometry.

Issue related to statistical analysis:

Overall, I do think the manuscript needs extensive english revisions. While I do find that the errors are minor in nature, there are a number of them throughout the manuscript that does detract from the work.

Response: We thank the reviewer for the suggestion. We have corrected the errors in this paper. Furthermore, we have purchased language editing services to revise the language used in this article. The editing certificate has been attached for your reference.

Round 2

Reviewer 1 Report

The corrected manuscript contains most of the Figures mentioned in the text, except Figure 7B (line 222). Also 7A is missing, but it is not mentioned in the text either.

General remarks:

Englesh needs revising, some sentences are not clear (marked in the text).

The abstract still does not contain all the most relevant information, especially on HPLC-QAMS method that was developed in the study.

HPLC-QAMS method and RCFs are still not sufficiently explained in the paper, and since it is one of the novelties of the study this part needs improving.

The explanation about the number of analyzed batched from method to method is not stated in the paper.

The quality of all the figures in the text is low and data is mainly unrecognizable.

The sample identification is not consistent within the text -  sometimes LBN, LBG and LBQ are used, sometimes the names of the provinces (Figures 2 and 3)...

The significance of the FTIR results is not clear - can they be used further in practice.

The numbers of some tables and figures in the text are mixted up.

Detailed comments are incorporated in the manuscript which is a part of this review.

Author Response

Issues related to figures and tables:

The corrected manuscript contains most of the Figures mentioned in the text, except Figure 7B (line 222). Also 7A is missing, but it is not mentioned in the text either.

Response: We thank the reviewer for this comment. We have corrected them in the text.

Figure 7A → Figure 6A

Figure 7B → Figure 6B

The quality of all the figures in the text is low and data is mainly unrecognizable.

Response: We thank the reviewer for this comment. We have improved the quality of all the figures, and adjusted the tables.

The sample identification is not consistent within the text - sometimes LBN, LBG and LBQ are used, sometimes the names of the provinces (Figures 2 and 3)...

Response: We thank the reviewer for this comment. We have used abbreviated forms like LBN, LBG, and LBQ in the text, and the meaning of these abbreviations is described below:

LBN: Lycium barbarum L. fruits from Ningxia

LBG: Lycium barbarum L. fruits from Gansu

LBQ: Lycium barbarum L. fruits from Qinghai

So, we used the names of the provinces to avoid repetition when captions of Figure 2 and Figure 3 mentioned the words “LB fruits from different regions”.

The numbers of some tables and figures in the text are mixted up.

Response: We thank the reviewer for this comment. We have checked and adjusted their positions in the text.

Issue related to English language:

English needs revising, some sentences are not clear (marked in the text).

Response: We thank the reviewer for this comment. We have corrected them according to your marks.

Issue related to abstract:

The abstract still does not contain all the most relevant information, especially on HPLC-QAMS method that was developed in the study.

Response: We thank the reviewer for this comment. We have checked the text and added the following point to the abstract:

“A new HPLC method was established for the simultaneous determination of 8 phenolic compounds by quantitative analysis of multiple components by a single marker (QAMS), including 4 phenolic acids (chlorogenic acid, caffeic acid, 4-hydroxycinnamic acid, and ferulic acid), 1 coumarin (scopoletin), and 3 flavonoids (kaempferol-3-O-rutinoside, rutin, and narcissoside). It was showned that rutin was the most dominant phenolic compound in LBQ, although the average content of 4 phenolic acids was also high in LBQ, and scopoletin was the richest in LBG. UHPLC-Q-TOF-MS was used to qualitatively analyze the phenolics, which showed LBN was abundant in phenolic acids, LBQ was rich in flavonoids, and coumarins were the most plentiful in LBG.”

Issue related to HPLC-QAMS method and RCFs:

HPLC-QAMS method and RCFs are still not sufficiently explained in the paper, and since it is one of the novelties of the study this part needs improving. 

Response: We thank the reviewer for the suggestion. We have enriched the explanation of HPLC-QAMS method and RCFs to “Introduction”:

“Within the scope of a certain linearity range, the amount (weight or concentration) of one component is proportionate to the response values of the detector.[29] Quantitative analysis of multiple components by a single marker (QAMS) is a simple and economical method that only requires one standard reference, and all analytes in the sample can be identified simultaneously.[30,31] It was essential to select a suitable internal reference (IR) in order to establish the relative correction factor (RCF) between IR and other effective ingredients, and RCF can be influenced by many factors, such as laboratories, chromatographic instrument systems, packing, and the models of chromatographic columns. [29,32] The method of QAMS has been widely accepted and applied in the quality control of herbal medicine, which has been adopted by the Chinese Pharmacopoeia, the United States Pharmacopoeia, and the European Pharmacopoeia Standards.[32]”

Issues related to analyzed samples:

The explanation about the number of analyzed batched from method to method is not stated in the paper.

Response: We thank the reviewer for this comment. We have explained the number of analyzed batched for “2.1” and “2.6” in “3.2. The appearance character of LB fruits”:

“The main physical characteristics of LB fruits from three different regions in China were observed. With the limitation of amounts of each batch of samples, three batches of samples were selected for each region because of their abundant quantities.”

Also, we have added the number of analyzed batched to “3.3. Extraction of phenolic compounds”:

“There were 38 batches of samples in total.”

Issue related to FTIR:

The significance of the FTIR results is not clear - can they be used further in practice.

Response: We thank the reviewer for this comment. We have added the following point to “2.3. FTIR-ATR”:

“Between them, the spectra of LBQ had two peaks at 2972 cm−1 (C-H stretching of the methylene bridges) and 2927 cm−1 (C-H stretching vabriation),[41-43] and the absorbance of LBG and LBN was only at 2928 cm−1 and 2927 cm−1, respectively. The absorption intensity of LBQ was higher than that of LBG and LBN at 2972 cm−1. A study identified 7 species and 3 variations of genus Lycium in China by FTIR, based on the additive infrared spectroscopy absorption of the chemical components and the differences of their relative contents in various Gouqi.[44] This method could provide a new way for the identification of LB fruits.”

Issues related to others:

Detailed comments are incorporated in the manuscript which is a part of this review.

Response: We thank the reviewer for this comment. We have corrected and improved them in the text according to your detailed comments in the manuscript.

Reviewer 2 Report

Images that were not previously attached to the manuscript are now of poor quality. Please correct it so that the entire manuscript can be evaluated.

Author Response

Issue related to figures:

Images that were not previously attached to the manuscript are now of poor quality. Please correct it so that the entire manuscript can be evaluated.

Response: We thank the reviewer for this comment. We have improved the quality of all the figures

Reviewer 3 Report

This is a review of a revised version of this manuscript. While I do thank the authors for the efforts in addressing many of the changes that I recommended, I am still not satisfied with the way that some of my comments have been addressed. 

Firstly, with the statements that I asked for clarification on, the authors need to make appropriate changes to the manuscript - for the following statements my question was answered but no changes were made to the manuscript to clarify (i.e. What does "LB is also the mainstream cultivation species in China.[3]" mean?  In Table 4, what is the criteria for a tick to be for the fruit from a region - does it need to be present in a single sample? All of the samples?) 

I am confused about Figure 5 - what does the red-blue scale of 2 to -2 mean? What is the representing? This should be better clarified. 

Looking at Figure 3, I find it very difficult to distinguish or see any obvious differences in the spectra for the fruits from different regions. I am not completely satisfied that there are obvious differences and that it is appropriate to say there is. 

Lastly, I am quite disappointed that the authors did not take on my suggestion to include some sort of multivariate analysis like PCA or PLS-DA in this work. If they were unable to see differences in the regions in this analysis, it surely would still give insight as to relationships between the studied compounds and their correlations with eachother and to the variability and diversity in the composition of the fruits. 

Author Response

Issues related to figures:

I am confused about Figure 5 - what does the red-blue scale of 2 to -2 mean? What is the representing? This should be better clarified. 

Response: We thank the reviewer for this suggestion. Heatmaps are generated by a hierarchical analysis of DEGs (y-axis) and individual samples (x-axis). Correlation matrix heatmap shows the correlation modules on the diagonal. Red represents the most positive Pearson correlation, and blue represents the most negative correlation.

We have added the classification to the caption of Figure 5:

“Heatmap of phenolic content in LB fruits from different regions. Red represents the most positive Pearson correlation, and blue represents the most negative correlation.”

Looking at Figure 3, I find it very difficult to distinguish or see any obvious differences in the spectra for the fruits from different regions. I am not completely satisfied that there are obvious differences and that it is appropriate to say there is.

Response: We thank the reviewer for this comment. We have corrected the expression in the text.

“2.3. FTIR-ATR

The results showed that there was some difference in the 3000-2800 cm−1 region.”

Issue related to the reference [3]:

What does "LB is also the mainstream cultivation species in China.[3]" mean?

Response: We thank the reviewer for this comment. There are seven species of Lycium in China. Among them, Lycium barbarum L. is the most widely distributed species, especially in the north and west of China. It has a strong ability to resist drought, salt-alkali, and low temperature, which can be as medicine and food to use in China, and has been recorded in the Chinese Pharmacopoeia. Nowadays, four provinces in China, including Ningxia, Hebei, Xinjiang, and Neimenggu, are the main areas of Lycium production, and the most cultivated species of Lycium is Lycium barbarum L..

We have modified the presentation of this sentence in the “Introduction”:

“LB is the most widely distributed cultivar in China.[3]” 

Issue related to Table 4:

In Table 4, what is the criteria for a tick to be for the fruit from a region - does it need to be present in a single sample? All of the samples?

Response: We thank the reviewer for this comment. We have added the explanation of the criteria for a tick to be for the fruit from a region in “3.2”:

“With the limitation of amounts of each batch of samples, three batches of samples were selected for each region because of their abundant quantities.”

The nine samples were representative integrating other experiments in the text. In table 4, it was showed the data which represented all of the samples from the same region.

Issue related to statistical analysis:

Lastly, I am quite disappointed that the authors did not take on my suggestion to include some sort of multivariate analysis like PCA or PLS-DA in this work. If they were unable to see differences in the regions in this analysis, it surely would still give insight as to relationships between the studied compounds and their correlations with each other and to the variability and diversity in the composition of the fruits. 

Response: We thank the reviewer for this suggestion. We have added PCA analysis to “2.6. Analysis of phenolic compounds in LB fruits by UPLC-Q-TOF-MS”:

“To obtain the overall characteristics and similarities of phenolic compounds in LB fruits from  three different regions, PCA test was performed. The results were shown that the LB fruits could be clearly differentiated into three groups in the PCA model (Figure 7).”

Figure 7. The PCA-X plot of LB fruits from different regions on the basis of 74 phenolic compounds identified by UPLC-Q-TOF-MS.

It is the basis of our ongoing study and the reason why we are going to develop quantitative mass spectrometry to explore LB fruits from different regions in our ongoing research.

Round 3

Reviewer 1 Report

This is the third reading of the manuscript and it has improved significantly, with more details, but also is more clear. The quality of some figures is improved.

Still, there are some issues that need improving and they are clearly stated in the pdf file that is a part of this review.

Most important issues are:

Figure 7 with the results of PCA analysis needs more discussion and explanation since it could be useful for origin determination.

There is a confusion in the used units in Table 3 and in connected discussion.

Some sentences are unnecessary repeated in the text.

Author Response

Review #1

Issue related to figures and tables:

There is a confusion in the used units in Table 3 and in connected discussion.

Response: We thank the reviewer for this comment. We have uniformed the unit (μg/g) that used in Table 3 and in connected discussion.

Issues related to repetition sentences:

Some sentences are unnecessary repeated in the text.

Response: We thank the reviewer for this comment. We have deleted them according to your marks.

Issue related to PCA analysis:

Figure 7 with the results of PCA analysis needs more discussion and explanation since it could be useful for origin determination.

Response: We thank the reviewer for this comment. We have enriched the discussion and explanation of PCA analysis to “2.6” (Page11):

“Principal component analysis (PCA) is a mathematical tool that aims to represent the variation present in the dataset using a small number of factors.[69] It is used to identify how one sample differs from another, which variables contribute most to the difference, and whether these variables are correlated.[70] Cossignani et al. found that the geographic origin of goji samples could be discriminated against using PCA for fatty acids and sterol percent compositions.[71] In a recent study by Gong et al., samples of Lycium barbarum L. from the same place could be partially discriminated by PCA using stable isotopes, earth elements, free amino acids, and saccharides.[72] To obtain the overall characteristics and similarities of phenolic compounds in LB fruits from three different regions, a PCA test based on identified 74 phenolic compounds was performed in this study. The two main principal components accounted for approximately 62.7% of the total variance. The results showed that in the PCA model (Figure 7), the LB fruits could be differentiated into three groups which contained LBN, LBQ, and LBG respectively.”

Reviewer 2 Report

The authors made important improvements in the article during the review process. Due to the importance of the analyzes performed in this work, I recommend it for publication.

Author Response

Thank you.

Reviewer 3 Report

This is my third review of this manuscript. The authors have made a better attempt at responding to many of my earlier feedback, particularly around addition of additional analyses. I still think aspects of my feedback and comments have not been appropriately addressed: 

Regarding the IR spectra - it is still very hard to see any difference in the spectra. Perhaps the additional peak at 2972 in LBQ is obscured such that this difference isnt obvious. The authors should either improve the figure so the difference is obvious, or need to amend their text more. 

For the heatmap - what do the authors mean regarding "positive Pearson correlation" - what is the correlation? Is this not a depiction of the relative abundance of each component in each of the samples? 

PCA - I think PCA would be better used for the quantitative HPLC data ( I think the authors have currently done with on the qualitative data? i.e. if the compound is present in the fruit?). Authors should include a better description and interpretation of the results also. 

Author Response

Review #3

Issues related to figures:

Regarding the IR spectra - it is still very hard to see any difference in the spectra. Perhaps the additional peak at 2972 in LBQ is obscured such that this difference isn’t obvious. The authors should either improve the figure so the difference is obvious, or need to amend their text more. 

Response: We thank the reviewer for this comment. We have improved the quality of Figure 3 in the text and enlarged the details of three curves that represented spectra of phenolic extracts in LB fruits from three different regions. So now, it is more clear to show the difference at 2972 cm−1 in LBN, LBG, and LBQ.

Figure 3. Spectra of phenolic extracts in LB fruits from different regions.

For the heatmap - what do the authors mean regarding "positive Pearson correlation" - what is the correlation? Is this not a depiction of the relative abundance of each component in each of the samples?

Response: We thank the reviewer for this comment.

Yes, it means the correlation. Heatmap is a two-dimensional visualization technique for high-dimensional data. In this graphical presentation of data, numerical values are displayed by colors. Each square indicates the Pearson correlation coefficient of a pair of compounds, and there is the possibility of positive and negative correlation. The value for the correlation coefficient is represented by the intensity of the blue or red color, as indicated on the color scale.

Issue related to PCA analysis:

PCA - I think PCA would be better used for the quantitative HPLC data ( I think the authors have currently done with on the qualitative data? i.e. if the compound is present in the fruit?). Authors should include a better description and interpretation of the results also. 

Response: We thank the reviewer for this suggestion.

In this study, we have established a new HPLC method for the simultaneous determination of 8 phenolic compounds by quantitative analysis of multiple components by a single marker (QAMS). The method was verified by samples of LB fruits from different regions, and the results showed that there were some differences in the contents of 8 phenolic compounds between them. However, there were overlaps between LB fruits from different regions in the following PCA model, and samples could not be discriminated entirely by PCA using 8 phenolic compounds.

A study from Jarouche et al. ( https://doi.org/10.3390/plants8120604) showed similar results, and a PCA test for the L. barbarum and L. chinense species samples indicated that they could not be differentiated based on 7 quantified phenolics tested by LC-MS. In the study of Ren et al. (https://doi.org/10.3390/molecules26175374), LB samples from three different geographical locations could be differentiated into two groups based on five quantified carotenoids, but there was highly overlapped between those LB fruits from Ningxia and Gansu in the PCA models.

So then, we performed a qualitative analysis by UPLC-Q-TOF-MS to determine the phenolic profile of LB fruits from different regions, and a PCA analysis has been used for the qualitative data. We have added the discussion and explanation of the PCA analysis to “2.6” (Page11):

“Principal component analysis (PCA) is a mathematical tool that aims to represent the variation present in the dataset using a small number of factors.[69] It is used to identify how one sample differs from another, which variables contribute most to the difference, and whether these variables are correlated.[70] Cossignani et al. found that the geographic origin of goji samples could be discriminated against using PCA for fatty acids and sterol percent compositions.[71] In a recent study by Gong et al., samples of Lycium barbarum L. from the same place could be partially discriminated by PCA using stable isotopes, earth elements, free amino acids, and saccharides.[72] To obtain the overall characteristics and similarities of phenolic compounds in LB fruits from three different regions, a PCA test based on identified 74 phenolic compounds was performed in this study. The two main principal components accounted for approximately 62.7% of the total variance. The results showed that in the PCA model (Figure 7), the LB fruits could be differentiated into three groups which contained LBN, LBQ, and LBG respectively.”

According to the results of qualitative analysis, we have planned to develop quantitative mass spectrometry to explore LB fruits from different regions in our ongoing research.
